# Using Unified Modeling Language to Analyze Business Processes in the Delivery of Child Health Services

**DOI:** 10.3390/ijerph192013456

**Published:** 2022-10-18

**Authors:** Fabrizio Pecoraro, Daniela Luzi

**Affiliations:** Institute for Research on Population and Social Policies, National Research Council, 00185 Rome, Italy

**Keywords:** business process modeling, UML, child care, child health, models of care, MOCHA

## Abstract

Business Process Management (BPM) has been increasingly used in recent years in the healthcare domain to analyze, optimize, harmonize and compare clinical and healthcare processes. The main aim of this methodology is to model the interactions between medical and organizational activities needed to deliver health services, measure their complexity, variability and deviations to improve the quality of care and its efficiency. Among the different tools, languages and notations developed in the decades, UML (Unified Modeling Language) represents a widely adopted technique to model, analyze and compare business processes in healthcare. We adopted its diagrams in the MOCHA project to compare the different ways of organizing, coordinating and delivering child care across 30 EU/EEA countries both from an organization and control-flow perspectives. This paper provides an overview of the main components used to represent the business process using UML diagrams, also highlighting how we customized them to capture the specificity of the healthcare domain taking into account that processes are reconstructed on the basis of country experts’ responses to questionnaires. The benefits of the application of this methodology are demonstrated by providing examples of comparing different aspects of child care.

## 1. Introduction

The management and provision of primary care services for children differ considerably from country to country [1], being developed over time according to socio-cultural and economic policies, and regulations that influence the types and ways healthcare services are provided. Embedded in health systems, which are complex organizations per se, they may encompass different aspects of care, privilege specific phases and/or conditions of child development, or implement particular solutions at either organizational or structural levels. Moreover, the evaluation of the achievements of the emerging guiding principles of quality of care [2] makes it necessary to analyze the different components that contribute to their fulfilment, suffice to think of quality principles, such as early-disease diagnosis, equity of access and integrated care. The evaluation of the degree of achievements, especially under the perspective of an inter-country comparison, requires the identification and analysis of a mix of interrelated aspects. They range, for instance, in the case of integrated care, from the coordination among health specialists to the one involving also social and/or school professionals, from the use of shared information tools to the planning and scheduling of common pathways of treatments. This means that along with the analysis of national policy statements, available resources and health outcomes, it is necessary to consider the procedural and organizational features implemented at country level and in real-life scenarios. To achieve this goal, we propose to introduce the Business Process Management (BPM) techniques [3,4] to complement and triangulate data that assess the quality of health systems. In this way, BPM contributes to consider aspects rarely analyzed within health processes [5,6], such as comparison across countries and across services, the organizational view that underscore the role and task distribution of human resources, the identification of complex KPIs (Key Performance Indicators), such as those indicating equity of access, continuity and integration of care that may be tackled in different parts of healthcare processes.

The aforementioned approach was successfully adopted in the MOCHA (Models of Child Health Appraised) project [7,8] that aimed to compare and appraise existing national models of primary care for children in 30 European Union (EU) and European Economic Area (EEA) countries. 

This paper intends to illustrate the methodology adopted to define, analyze and compare the various pathways of child care detected in the MOCHA project as part of the multidimensional analysis carried out within the project. It also intends to show, on the basis of selected examples (for complete use cases see [1,9,10,11,12]), how UML (Unified Modeling Language) can support cross-country comparison with a special focus on pattern of collaboration as well as the identification of complex KPIs. It first provides an overview of studies published within the BPM framework focusing the attention on the healthcare domain. The subsequent section describes the procedures of data collection to define and compare existing models of primary care for children in Europe within the MOCHA project. It also provides an overview of the methodologies adopted to model business processes highlighting languages and diagrams specifically adopted and customized to describe the behavior and the structure of child health systems. The methodological section finally describes the different steps carried out to perform the cross-country comparison. Some examples on the different ways of comparing diverse aspects of child care are provided so as to show the benefits of the application of this method which are then summarized in the discussion and conclusions.

## 2. Related Work

This paragraph examines previous BPM studies which deal with healthcare process comparison, privilege the organizational perspectives and include KPIs in the process analysis. Within the vast variety of BPM literature [13], this restriction helps in identifying the major challenges of process analysis in the healthcare domain [14] and, at the same time, motivate the methodological choices of our approach. As an increasing number of studies, especially in the healthcare domain, used process mining techniques to analyze BPM, we mainly consider these studies as a reference point for the analysis of related work [6,15,16,17,18].

Process comparison is generally carried out to reconstruct the AS-IS sequence of activities against a to-be model, usually a clinical guideline or medical protocol, to identify correspondences and deviations that may lead to the improvement of the actual process or to the update of the clinical prescriptive model. For this reason, some authors refer to this type of comparison as a conformance checking or conformance analysis [6,17,19]. These studies [20,21,22] often include and overlap with a consistent number of papers that aim to identify variants and similarities of pathways within a cohort of patients with similar health conditions [23,24,25,26].

Moreover, the identification of process variability in the treatment of specific pathologies/medical conditions (e.g., cancer, stroke) and/or in a clinical setting (i.e., emergency room, surgery) is usually hospital-based and rarely applied to analyze cross-organizational processes. Exceptions concern the identification of the variability of the patient flows in an emergency department (ED) and in the ward of four Australian hospitals [27,28]. In this study, the authors adopted Petri Nets to represent each discovered process and considered time variation in terms of waiting time and length of stay. Andrews and colleagues (Andrews et al., 2016) proposed a static and dynamic comparison to trace two cohorts of patient flow in an ED of two Australian hospitals using a visualization component based on a colored Business Process Model and Notation (BPMN) representation. Leonardi et al. [29] performed a process discovery analysis of four Italian hospitals treating ischemic stroke and defined a mechanism for abstracting event log traces based on ontology that, hiding unnecessary details of the process, facilitates the identification of relevant, high-level differences/variations between processes as well as clustering techniques.

A few studies considered health processes outside hospitals and/or in conjunction with care pathways carried out in other health services, such as those available in primary care. Mans et al. [30] applied process mining to analyze the procedures for treating patients with stroke in different hospitals connecting it to a pre-hospital dataset gathered through patients’ interviews. In this way, they could trace the temporal information about the action taken by patients, parents and their GP (General Practitioner) before the admission and the timestamps for diagnostic and treatment activities. More recently, Sato and colleagues [31] correlated treatment processes of ischemic stroke patients with primary care data available on a population-based stroke database, thus helping the improvement of treatment flows as well as prevention actions.

In our approach, clinical guidelines were a reference point that guided the construction of questionnaires used to collect information on clinical and organizational procedures put in place in specific child health tracer conditions at the national level. The results of the questionnaire provided us with data for the process comparison across countries at a high level of description. This also allowed us to analyze care pathways across different health services, such as primary and secondary care, as well as to compare processes in different tracer conditions. These types of comparison were rarely performed in the above-mentioned studies.

While clinical processes are generally the major focus of process analysis and discovery, studies that consider the organizational perfective are fewer in number and rely on a mix of methods (i.e., workflow models and social network analysis) to reconstruct the use of resources, especially when considering health professionals. These studies reconstruct interaction models and handover of task among professionals in specific settings [32,33,34]. Conca and colleagues [35] identified seven collaboration patterns in the treatment of type 2 diabetes mellitus and compared these patterns with the clinical evolution of the patients within the context of primary care.

Time-related aspects (such as waiting time, time spent per task/activity and length of stay) of both the control follow and the organizational perspective are the most frequently used KPIs to measure the performance of the processes. Some works related personnel handover interaction with the time to perform a medical or a managerial activity [36,37,38]. Grando and colleagues [39] applied observational techniques (interviews and video ethnography) and process mining to analyze personnel’s sequential interactions in a preoperative setting to deduce time spent with patient groups including time spent to use electronic information systems. 

In our approach, the organizational perspective had a crucial importance and was specifically focused on the health professionals’ roles which were analyzed in terms of collaboration and team composition in the performance of both clinical and managerial activities. This also motivated the choice of a modeling language that addresses the organizational components of the process and is flexible enough to allow cross-country comparison. Moreover, considering KPIs, it was possible to compare time-related aspects of the control flow considering the straightforwardness of the sequence of activities performed at a national level as indicators of possible bottlenecks and waiting times. In addition, and more importantly, our approach made it possible to analyze composite quality indicators, such as practices towards integrated care, which can be detected considering different aspects and specific sub-processes. The identification of similar sub-processes in different child tracer conditions allowed us to analyze whether and in which country these improvements towards key principles of healthcare systems, such as equity of care continuity and integrated care, were put in place as well as their variability.

## 3. Materials and Methods

### 3.1. Data Collection

In order to compare and appraise existing national models of primary care for children in Europe, the MOCHA project appointed experts (CA, Country Agent) in the field of child care in 30 countries. Each CA was required to directly answer on a wide range of questionnaires or to involve additional experts depending on their specific expertise to give a country’s unique perspective. Questionnaires were prepared by the different research teams involved in the project focusing on a specific aspect of child care, and each of them was reviewed both by the management team and by the MOCHA External Advisory Board before being sent to each Country Agent for answers [40].

For the purpose of reconstructing the process in place in each country for the delivery of child health services, questionnaires have been based on specific scenarios to gather information on the management and treatment of children considering different tracer conditions at various children’s life stages, such as mental health, asthma, LTV (Long-Term Ventilation), and TBI (Traumatic Brain Injury). These case studies and the relevant questions have been designed to set the criteria of comparable process descriptions specifying the macro-processes that describe the scenario’s generic workflow and related macro-activities. This has been used to single out homogeneous, comparable parts of the process that have to be taken into account when analyzing the different national care provisions with the purpose of identifying specific parts of the care and management process, such as diagnosis, access to care services, and continuity of care.

### 3.2. Choice of the Modeling Language

Different tools and techniques have been proposed in the literature to model, implement, and execute business processes as well as to refine them based on clinical and administrative information [14,41,42]. Among them, BPMN and the UML activity diagram are widely adopted to model business processes [43,44,45,46,47], as well as in complex systems such as healthcare [48,49,50,51,52].

Both aforementioned languages describe the process flow highlighting its activities as well as the resources and the actors involved in their execution [47,53]. These two alternatives were compared over years from different perspectives and considering different evaluation criteria, e.g., [53], such as diagram understandability by readers, adequacy to represent business processes from a graphical perspective, and easiness in mapping the diagrams to an executable language. Generally, BPMN and UML activity diagram are considered similar in their potential of representing the business process behavioral perspective. BPMN tends to prevail over UML when associated with simulation as it can be automatically mapped to a Business Process Execution Language (BPEL). However, one of the main limits of BMPN is that it does not model the static part of the system and, in particular, the organizational perspective [54] which are instead well described in the UML use case diagrams. These types of diagrams describe both the behavior (dynamic comportment) and the structure (static description) of the process under different perspectives and levels of abstraction. Moreover, the flexibility of UML guarantees an easy update of its diagrams during the development of the process [55] as well as the customization and extension of its models to target specific needs and capture the characteristics of particular settings. Finally, UML diagrams not only are easily usable by non-IT stakeholders [56,57,58,59], but also provide an easy-to-read specification and documentation of the process from the end-users’ perspective [60]. This is a crucial point for the purpose of our analysis as UML had to support healthcare professionals, policy makers, and other stakeholders to easily interpret the main organizational differences between countries in the provision of child care, as well as to identify crucial points of the process that need improvements to enhance the quality of care.

### 3.3. UML Customization

Some elements and notations of the UML diagrams had to be extended and customized to capture the specificity of the business process modeling in order to perform cross-country comparison and document CAs’ responses to the questionnaires. In particular, extension mechanisms have been applied to model two main aspects that are not considered in the current UML behavioral diagrams: (1) healthcare professionals can be involved in the different steps of a child’s care pathway as part of a team; (2) a single activity can be performed by multiple actors (i.e., joint activity) contributing with different roles and responsibilities. To address these issues, the following notations have been introduced and adopted in our methodology: Organizational perspective: the aggregation relationship (
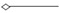
) adopted in the UML class diagram to associate two objects of the system in a part-whole or part-of relationship has been introduced in our methodology to aggregate different types of professionals in a team that is involved as a whole in a specific use case (left side of Figure 1). This is crucial, considering, for instance, that health and social care professionals can manage a child’s pathway either working as individuals carrying out specific professional-related activities or working in close collaboration in a fully integrated system as part of a multi-disciplinary team. In the example reported below (left side of Figure 1), the activity of developing a written plan is performed by a secondary care physician (SC) in Italy, by a team of professionals belonging to the secondary care (SC Team) setting in Romania, and by a primary care physician (PC) in Lithuania, while in Malta, professionals belonging/part of/the primary and secondary care work in a team (PC–SC Team) to accomplish this task. This aspect is also described in the activity diagram representing the PC–SC Team using an overall swim lane that can be subsequently divided into swim lanes to list the activities performed by each professional that composes the team (right side of Figure 1). This makes it possible to capture not only the presence of a multidisciplinary team, but also to detail the different types of professionals that compose the team and their roles.

Control-flow perspective: each action that is simultaneously performed by multiple actors is duplicated in the relevant swim lane of the activity diagram and encapsulated within fork and join elements (right part of Figure 1). This makes it possible to represent parallel activities performed by different professionals, thus mirroring the relationship described in the use case. This is shown in the example reported in the right hand of Figure 1, where the development of the personalized written plan is executed jointly by the Neurologist and the PC physician as part of a Team composed by both PC and SC physicians (PC–SC Team).

Moreover, as already highlighted in Figure 1, an additional adaptation of the UML notations was used to include specific notes related to the elements of the use case diagram: green notes capture the type of actor (s) that are involved in the related macro-activity at a country level. This provides a first snapshot of the different national organization features as reported by the CAs. For instance, considering the development of the written plan shown in Figure 1, the Primary Care professional (PC) is involved in Lithuania, the Secondary Care professional (SC) is involved in Italy, while a PC–SC Team in involved in Malta. Similarly, red notes can be used to specify which is the question analyzed to capture professionals involved in the relevant use case. This was important not only to track and, if needed, to verify the results of the questionnaire analysis, but also to compare responses related to similar topics present in different questionnaires (e.g., use of a personalized plan in Long-Term Ventilation (LTV), Traumatic Brain Injury (TBI), and intractable epilepsy).

### 3.4. Methodological Steps

The methodology adopted in the MOCHA project to model and analyze business processes in the healthcare context is reported in Figure 2, highlighting the main steps to be accomplished to define the UML use case and activity diagrams that describe both the organizational and the control flow views as well as the sources of information adopted to accomplish these tasks.

The first part of the methodology concerns the definition of an aggregated use case diagram that lists the main macro-activities to be analyzed. Each use case describes a specific part of the care process as well as the actors who are involved in the performance of these macro-activities. The aggregation relationship introduced allows also to capture the professional’s level of collaboration in the performance of the different tasks. This use case diagram is firstly initialized considering the CA questionnaires and cyclically updated on the basis of the answers reported by CAs. In addition, relevant guidelines, scientific articles and/or documentation sent by CAs can be analyzed to enrich the description of the care pathway and to capture relevant use cases to be included in the model. For each use case, green and red notes are used to specify, respectively, which are the professionals involved for each country and the relevant question analyzed. Once all questionnaires sent by the CAs are analyzed and the use case diagram is completed, a snapshot of the macro-activities was provided along with a detailed list of actors performing them in each country. Based on these results and depending on the research target question, different types of analysis can be performed to group countries with similar behavior and organizations in the delivery of care.

Once the use case diagram is defined, the next step of the methodology is the production of activity diagrams that further detail, for each identified group of countries, the chronological order, the triggering conditions as well as possible waiting times. The comparison between groups of countries can be also quantified associating business process metrics [61,62] with the elements of the activity diagram to assess the efficiency of the process under investigation. 

## 4. Results

The methodology proposed in this paper has been the basis to analyze different scenarios and tracer conditions in the MOCHA project [9], such as mental health [10], asthma [11], LTV [1], and TBI [12]. Hereafter, this approach is described by giving examples that illustrate how UML has been applied to represent parts of the processes pointing out the different perspectives they can represent. From the organizational perspective, we provide an example focused on the level of collaboration of health, social, and school professionals in the joint implementation of the personalized written plan, which is considered one of the indicators of care integration. The second example is based on the control flow that describes the procedures adopted by a primary care professionals to refer the child to a specialist. This is part of a wider process that describes the interface between the model of primary and secondary/hospital care, and is considered one of the indicators related to the continuity of care.

### 4.1. Organizational View

This is an example of longitudinal analysis that compares the team composition in two complex care scenarios across countries. This analysis contributes to identifying whether and how one of the indicators of the level of integrated care has been actualized and whether this is a common organizational feature achieved in different tracer conditions. Moreover, it contributes to providing further insight on the team composition detailing the type of professionals involved as an indicator of horizonal and/or vertical integration of care. 

Figure 3 shows the UML use case diagram related to the implementation of the personalized written care plan for an adolescent with TBI and children on LTV. There is a high variety of team composition across countries and also between health conditions. The wider level of team composition, comprising PC, SC, Social Care (SoC), and School Care (ShC) professionals is present in a limited number of countries (Denmark, Ireland, the Netherlands, and Norway) in the treatment of TBI, while in the treatment of LTV, the same countries, with the exception of Denmark, comprises the collaboration of professionals from the primary and the secondary care settings. The presence of other types of teams is higher in TBI than in LTV in eight countries (Denmark, Estonia, Finland, Ireland, Lithuania, the Netherlands, Norway, and Romania). In three countries, the written plan is implemented by individual professionals both in TBI and LTV (Croatia, Hungary, Portugal).

Table 1 provides an overview of the results reported in the use case diagram summarizing the classification of the team composition that implements the personalized plan in both complex care conditions.

### 4.2. Control-Flow View

The example reported below describes how UML models the interaction between primary and secondary care professionals in the treatment of children with asthma in case of an exacerbation. It considers two activities: the referral to a specialist and the communication of results, in this case, the ones related to the spirometry test. This analysis contributes to examining the level of care coordination across countries from both sides of the primary/secondary interface as a key indicator to assess the fragmentation of service provision or, otherwise, the efforts to the promotion of continuity of care.

Figure 4 shows the use case diagram related to the referral as well as to the communication procedures as part of the work presented in [9]. In particular, considering the referral procedure, all CAs have reported that the PC prescribes the visit and refers the child to a specialist. Differences among countries are found considering the booking procedures. In particular, in five countries (Belgium, Croatia, Finland, Portugal, Spain), the primary care physician directly chooses the specialist and books the visit, while in another five countries (Bulgaria, France, Lithuania, Malta, and the Netherlands), these activities are performed by the family. In the remaining countries, either parents or the GP/pediatrician can choose the professional and book the visit depending on the level of collaboration among professionals (also co-location). Alternatively, parents can decide to choose the specialist and book the visit. Considering the communication of the spirometry results, the specialist reports to the primary care professional either via a direct channel between professionals or involving the parents as conduits of the transmission of results. As highlighted by the green note, in nine countries, there is a direct communication among professionals using a shared EHR (Finland, Spain), a letter (Croatia, Germany, Iceland, Ireland, the Netherlands, Romania) or a report (Cyprus), while in three countries (Bulgaria, Latvia, Lithuania), this task is conducted by parents who generally have a letter containing the results of the specialist’s visit. In the remaining countries, the communication can be conducted by parents or by the specialist depending on the availability of a shared record within the specific region, or on the asthma and exacerbation severity as assessed by the specialist.

Starting from the results reported in the use case diagram, two countries have been analyzed and modeled using the UML activity diagram (Figure 5) to capture the level of complexity of the process in terms of its straightforwardness in the primary care–secondary care collaboration/interaction procedures. On the left side (Spain), both activities are directly performed by the health professionals, taking advantage of the availability of a shared electronic health record (EHR swim lane) with no further burden for the parents. On the right side (Lithuania), the parents are involved both in the referral and in the communication procedures, implying certain possible wait time actions represented by the hourglass and a full involvement of the parents in the organization of the child pathway (to choose and book the specialist’s visit as well as to communicate the spirometry results to the PC physician).

## 5. Conclusions

This paper describes a novel application of UML to compare and evaluate national health systems. In our approach, it provides an additional performance perspective on how national healthcare policies and regulations are translated into practice on the basis of local available resources and medical and organizational procedures. The adoption of UML facilitates the representation of the different ways of organizing and delivering child care and helps the extraction of uniform parts of the healthcare process, as a prerequisite for a cross-country comparison. Moreover, the real-life scenarios derived from ad hoc questionnaires made it also possible to analyze process variability among different tracer conditions at both national and cross-country level. The possibility to compare an entire child pathway or part of it and/or focusing on specific aspects of care (e.g., presence of teams across similar health conditions; use of ICT in different parts of the process) helps highlighting the mix of components that are variously combined to implement national models of child care. This provides an additional view that contributes, along with the other analysis carried out in MOCHA, to gain insights of the different efforts necessary to implement optimal models of child healthcare, under a multidimensional perspective.

These types of comparisons are not possible using other approaches widely diffused in the literature, such as process mining, as the data acquisition would require, at least, a hard to achieve harmonization and integration of different information systems both at the local and cross-national level. Of course, our approach has the limit of providing a high-level process description detected from questionnaires which could be biased by respondents’ subjective views and analysts’ interpretations. These limitations were partially overcome through the analysis of national guidelines and additional documentation. Moreover, the pictorial description of the process derived from the analysis was sent back to CAs to check its conformance. Particularly important, in this perspective, is the easiness of reading guaranteed by the use of UML diagrams that support healthcare professionals, policy makers, and other stakeholders to easily interpret the main organizational differences between countries in the provision of child care, as well as to identify crucial points of the process that need improvements to enhance the quality of care.

Considering, in particular, the organizational perspective, our mayor efforts were focused on the identification of patterns of collaboration and team composition in the delivery of child care. In our view, this feature, which is difficult to detect using other approaches and rarely analyzed in the literature, can be considered among the KPIs that denote optimal models of child care. In fact, principles such as equity, integrated care, and continuity of care can be achieved in different ways, supporting the coordination among health professionals, involving the child and parents in the planning and scheduling of a personalized treatment, using shared Electronic Health Records (EHRs) to monitor health status, just to mention a few. The detection of these features in different pathways and tracer conditions can be a valuable addition to the KPIs generally considered to evaluate the care process.

Moreover, the analyses of process performance could be improved by associating activities, interactions, decision points, and waiting times with other quality indicators such as health outcomes, expenditure for health professionals, or avoidable hospitalization, so as to analyze the effectiveness of the process under investigation. Of course, this needs the availability of robust and reliable data—which are often lacking, especially in relation to child healthcare [63]—as well as the achievements of experts’ consensus on the aspects and related measures to be considered and collected to capture optimal pathways of child care.

## Figures and Tables

**Figure 1 ijerph-19-13456-f001:**
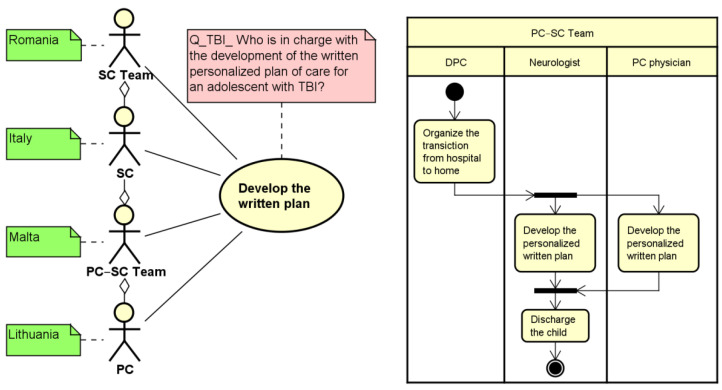
An example of customization of use case (left side) and activity (right side) diagrams, for Developing a Personalized Care Plan. PC = Primary care professionals; SC = Secondary care; DPC = Discharge Planning Coordinator.

**Figure 2 ijerph-19-13456-f002:**
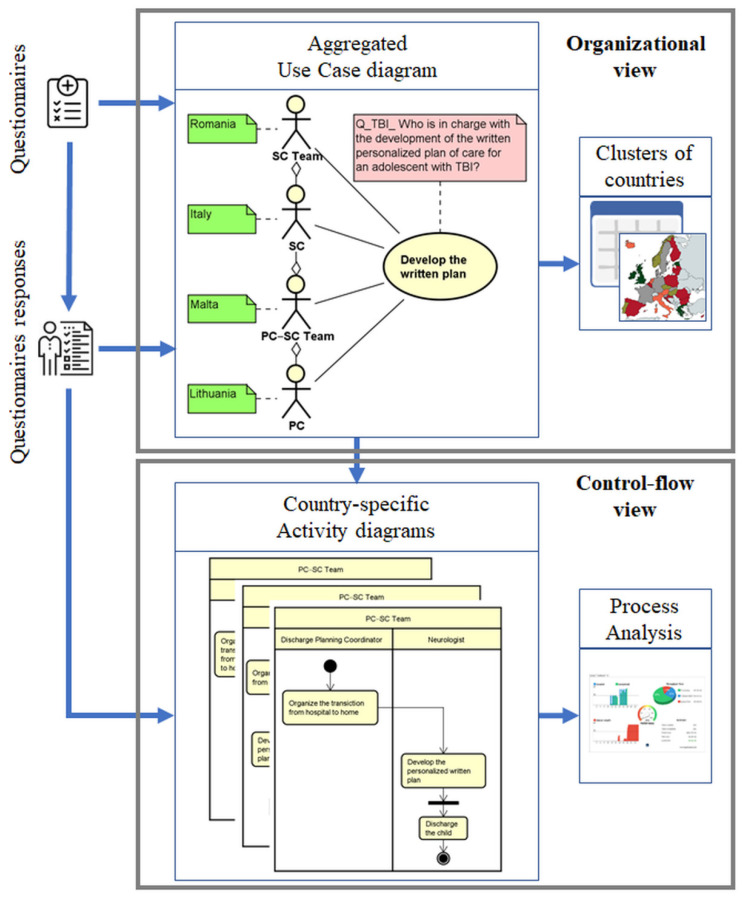
Steps of the methodology highlighting the input, the UML diagrams, and the analysis performed in the organization and in the control-flow views.

**Figure 3 ijerph-19-13456-f003:**
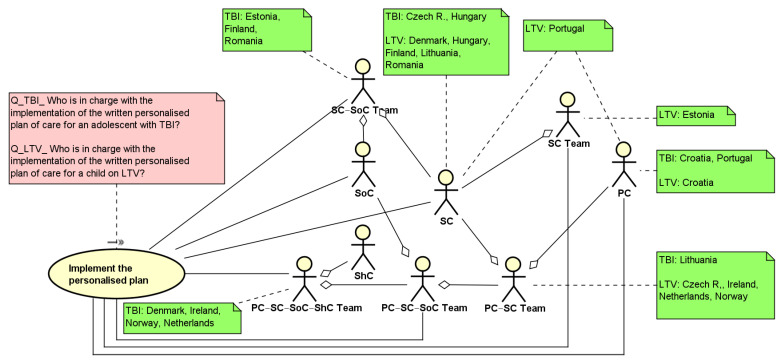
UML use case diagram describing the macro-activity: Implement the personalized plan. PC = Primary care professionals; SC = Secondary care; SoC = Social care; ShC = School care.

**Figure 4 ijerph-19-13456-f004:**
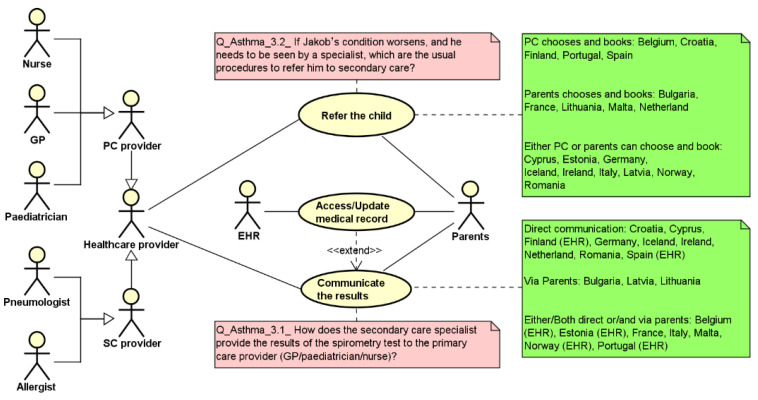
Use case diagram highlighting the responses reported by the CAs (green notes) for the specified processes (red notes).

**Figure 5 ijerph-19-13456-f005:**
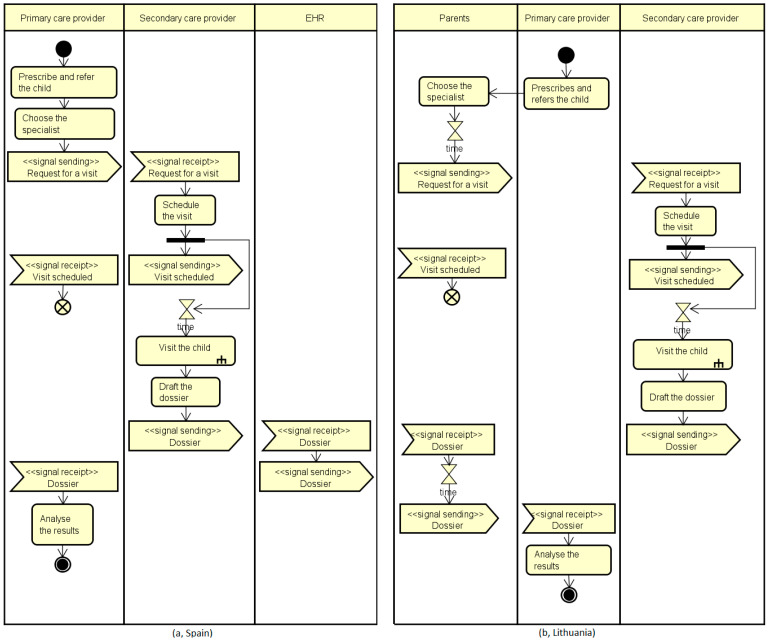
Activity diagrams reporting two control flows: on the left side (Spain), both activities are directly performed by the health professionals, while on the right side (Lithuania), the parents are involved both in the referral and in the communication procedures.

**Table 1 ijerph-19-13456-t001:** Cluster of countries considering the implementation of the personalized written plan for adolescents with TBI and children on LTV.

		LTV				
		PC–SC–Soc–ShC Team	SC–SoC Team	PC–SC Team	SC Team	Individual Professional (s)
TBI	PC–SC–Soc–ShC Team			Ireland Norway Netherlands		Denmark
	SC–SoC Team				Estonia	Finland Romania
	PC–SC Team					Lithuania
	SC Team					
	Individual professional(s)			Czech R.		Croatia Hungary Portugal

## Data Availability

Not applicable.

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
