# Peer review of "Using Unified Modeling Language to Analyze Business Processes in the Delivery of Child Health Services"

_ijerph, 2022, doi:10.3390/ijerph192013456_

Round 1
Reviewer 1 Report
1. Since the issue of the use of UML, in various fields, is not a very topical one, I recommend including in the paper a Literature Review section, in which the opinions of other authors who have published reference papers regarding this subject are presented, aspect that would highlight the foundation of the research that the authors undertook.
2. I believe that within section 2.1. Business process discovery, the description of the interaction between the data collection process, the IT systems used, the Data Mining processes (in the context of the MOCHA project) and the use of UML is extremely brief by the authors, an aspect that complicates the process of understanding the research objective and of the results obtained. In addition, the data that formed the basis of the analysis performed by the authors must be presented, with great accuracy, to facilitate the efficiency of the method proposed by the authors.
3. I think that in section 3. Discussion and Conclusions it would be appropriate to insist on the detailed presentation of the effectiveness of the method proposed by the authors, as well as conclusions of the research undertaken. As they are currently presented, they are difficult for readers to understand/anticipate and reduce the scientific value of the paper.
Author Response
Reviewer's comment: Since the issue of the use of UML, in various fields, is not a very topical one, I recommend including in the paper a Literature Review section, in which the opinions of other authors who have published reference papers regarding this subject are presented, aspect that would highlight the foundation of the research that the authors undertook.
Authors' reply: As suggested we added two paragraphs: the first one (2. Related work) examines previous business process modelling studies which deal with healthcare process comparison, while the second one (2.2. Choice of the modelling language) analyses the different tools proposed in the literature for business process modelling. It also reports the main advantages and issues in using UML for the scope of our study.
Reviewer's comment: I believe that within section 2.1. Business process discovery, the description of the interaction between the data collection process, the IT systems used, the Data Mining processes (in the context of the MOCHA project) and the use of UML is extremely brief by the authors, an aspect that complicates the process of understanding the research objective and of the results obtained. In addition, the data that formed the basis of the analysis performed by the authors must be presented, with great accuracy, to facilitate the efficiency of the method proposed by the authors.
Authors' reply: We updated section 2.1 highlighting the procedures adopted to collect data at country level. As reported, our methodology relies on a set of scenario-based questionnaires sent to experts in the field of child care. Each questionnaire aimed to collect information on the policies and procedures in place in each country to compare and appraise existing national models of primary care for children in Europe.
Reviewer's comment: I think that in section 3. Discussion and Conclusions it would be appropriate to insist on the detailed presentation of the effectiveness of the method proposed by the authors, as well as conclusions of the research undertaken. As they are currently presented, they are difficult for readers to understand/anticipate and reduce the scientific value of the paper.
Author's reply: Discussion and conclusions section has been updated to detail the main purpose of our methodology also highlighting pros and cons in the adoption of UML to model business processes to compare and appraise primary care models in Europe.
Reviewer 2 Report
The article has little value from the perspective of contributing to the development of science. It is not known what hypothesis the authors wanted to confirm. The text lacks a description of precise criteria that allow to evaluate the advantages and disadvantages of UML in comparison with other techniques. It was also necessary to describe the limitations of the applied research approach but there is nothing like this in the article... The "discussion" section is practically non-existent. The article can only be classified as an imprecise case study.
Author Response
Reviewer's comment: The article has little value from the perspective of contributing to the development of science. It is not known what hypothesis the authors wanted to confirm. The text lacks a description of precise criteria that allow to evaluate the advantages and disadvantages of UML in comparison with other techniques. It was also necessary to describe the limitations of the applied research approach but there is nothing like this in the article... The "discussion" section is practically non-existent. The article can only be classified as an imprecise case study.
Authors' reply: The paper has been improved on the basis of the reviewers’ suggestions. In particular, we added two paragraphs: 1) Related work that examines previous business process modelling studies which deal with healthcare process comparison; 2) Choice of the modelling language that analyses the different tools proposed in the literature for business process modelling. It also reports the main advantages and issues in using UML for the scope of our study. We improved the materials and methods section to better specify which are the source of data and the methodology adopted to model business processes from an organizational and control-flow perspectives. Moreover, discussion and conclusions section has been updated to detail the main purpose of our methodology also highlighting pros and cons in the adoption of UML to model business processes to compare and appraise primary care models in Europe.
Round 2
Reviewer 1 Report
Accept in present form
Reviewer 2 Report
Accept in present form.